# Efficient Induction of Antigen-Specific CD8^+^ T-Cell Responses by Cationic Peptide-Based mRNA Nanoparticles

**DOI:** 10.3390/pharmaceutics14071387

**Published:** 2022-06-30

**Authors:** Sigrid D’haese, Thessa Laeremans, Sabine den Roover, Sabine D. Allard, Guido Vanham, Joeri L. Aerts

**Affiliations:** 1Laboratory for Neuro-Aging and Viro-Immunotherapy (NAVI), Faculty of Pharmacy and Medicine, Vrije Universiteit Brussel, 1090 Brussels, Belgium; sigrid.katrien.dhaese@vub.be (S.D.); thessa.laeremans@vub.be (T.L.); sabine.den.roover@vub.be (S.d.R.); 2Department of Internal Medicine (IRG), Vrije Universiteit Brussel, Universitair Ziekenhuis Brussel, 1090 Brussels, Belgium; sabine.allard@uzbrussel.be; 3Department of Virology, Institute of Tropical Medicine, University of Antwerp, 2000 Antwerp, Belgium; gvanham@ext.itg.be

**Keywords:** mRNA, nanoparticles, nanovaccines

## Abstract

A major determinant for the success of mRNA-based vaccines is the composition of the nanoparticles (NPs) used for formulation and delivery. Cationic peptides represent interesting candidate carriers for mRNA, since they have been shown to efficiently deliver nucleic acids to eukaryotic cells. mRNA NPs based on arginine-rich peptides have previously been demonstrated to induce potent antigen-specific CD8^+^ T-cell responses. We therefore compared the histidine-rich amphipathic peptide LAH4-L1 (KKALLAHALHLLALLALHLAHALKKA) to the fully substituted arginine variant (LAH4-L1R) for their capacity to formulate mRNA and transfect dendritic cells (DCs). Although both peptides encapsulated mRNA to the same extent, and showed excellent uptake in DCs, the gene expression level was significantly higher for LAH4-L1. The LAH4-L1–mRNA NPs also resulted in enhanced antigen presentation in the context of MHC I compared to LAH4-L1R in primary murine CD103^+^ DCs. Both peptides induced DC maturation and inflammasome activation. Subsequent ex vivo stimulation of OT-I splenocytes with transfected CD103^+^ DCs resulted in a high proportion of polyfunctional CD8^+^ T cells for both peptides. In addition, in vivo immunization with LAH4-L1 or LAH4-L1R–mRNA NPs resulted in proliferation of antigen-specific T cells. In conclusion, although LAH4-L1 outperformed LAH4-L1R in terms of transfection efficiency, the immune stimulation ex vivo and in vivo was equally efficient.

## 1. Introduction

The delivery of nucleic acids to eukaryotic cells has been extensively investigated for gene therapy as well as for vaccination purposes [1,2,3]. For both applications, it is crucial that the formulation increases cellular uptake and ensures endosomal escape [4]. In the context of vaccine research, the encoded antigen should be presented by MHC molecules on antigen-presenting cells in combination with co-stimulatory molecules and secretion of cytokines. Several formulation strategies have been investigated for efficient delivery of nucleic acids [5], including polymers [6], cationic peptides [7], metal ions, inorganic particles [8,9], and lipid formulations [10].

Although cationic peptides have mainly been investigated for gene therapy applications, their use in vaccine formulations is promising. Usually, in addition to apolar amino residues and histidine, the peptides contain lysine and/or arginine. The cationic nature of the peptides allows efficient complexation to the negatively charged nucleic acids. One of the first cationic peptides described for DNA delivery was KALA (N-WEAKAKALARALAKHLAKALAKALKACEA-C), which resulted in efficient transfection of plasmid DNA (pDNA) [11]. However, arginine is present in naturally occurring DNA-binding motifs and was suggested to be less toxic. Moreover, poly-L-arginine polymers outperformed poly-L-lysine polymers in pDNA transfection, suggesting that arginine-based transfection reagents are more efficient [12]. For these reasons, lysine was substituted with arginine to create RALA. RALA efficiently delivered pDNA in vitro as well as in vivo [13]. Furthermore, RALA–mRNA complexes induced CD8^+^ T cells’ responses towards the model antigen ovalbumin (OVA) [14].

Recently, with the purpose of increasing RALA’s pDNA transfection efficiency, HALA peptides were developed. The largest difference compared to RALA is the presence of four histidine residues instead of one, resulting in a histidine-rich peptide (HRP). Unfortunately, the increase in histidine residues in HALA peptides did not significantly improve transfection efficiency across all cell lines tested [15]. Nevertheless, comparison between RALA and LAH4-L1 (KKALLAHALHLLALLALHLAHALKKA)—another HRP containing four histidine residues—showed that LAH4-L1 outperformed RALA in transfecting mRNA in dendritic cells (DCs) [16]. To further improve the delivery, LAH4-L1–mRNA complexes were added to polylactic acid (PLA) lipoparticles. Although it was shown that in transfected monocyte-derived DCs, maturation levels were increased and inflammatory cytokines were produced, the possibility cannot be excluded that this was due to an mRNA-related effect [16]. Interestingly, LAH4 complexing OVA peptides in combination with the adjuvant CpG was able to induce a stronger CD8^+^ T-cell response than OVA–CpG alone [17]. These results indicate that LAH4-L1 has a potential adjuvant effect, but this was not studied in detail.

The combination of cationic and histidine residues in LAH4 allows for membrane disruption in bacteria, causing lysis, and giving rise to powerful antimicrobial peptides [18,19,20]. A well-known naturally occurring member of this class of peptides is LL-37, or cathelicidin. LL-37 is released by neutrophils in an inflammatory setting, and has been shown to be able to activate the nucleotide-binding and oligomerization domain-like receptor family pyrin domain containing 3 (NLRP3) inflammasome after lipopolysaccharide (LPS) priming in murine myeloid cells [21]. Activation of the sensor NLRP3 leads to assembly of the NLRP3 inflammasome, resulting in the secretion of inflammatory cytokines such as IL-1β [22]. IL-1β has been shown to increase expression of the co-stimulatory molecules CD40, CD80, and CD86 in DCs, as well as to enhance antigen presentation in the context of both MHC I and MHC II [23]. Moreover, IL-1β enhances the expansion of CD4^+^ T cells through increased proliferation as well as improved survival [24], and plays a role in the induction of polyfunctional CD8^+^ T cells [25,26].

Therefore, we aimed to compare two HRPs containing lysine or arginine residues for their ability to complex mRNA and protect it from ambient RNases. To assess transfection efficiency in vitro, we used a more relevant DC model, namely, migratory CD103^+^ DCs. This DC subset has been shown to be the DC subset responsible for cross-presentation of antigen and CD8^+^ T-cell stimulation [27]. Furthermore, we assessed the ability of the HRPs to induce inflammasome activation. Lastly, we assessed the capacity of transfected DCs to induce polyfunctional CD8^+^ T-cell responses in vitro and proliferative CD8^+^ T-cell responses in vivo.

## 2. Materials and Methods

### 2.1. Mice

CD45.1 mice were housed at the breeding facility of the VUB (Brussels, Belgium), and were between 6 and 12 weeks of age at the start of the experiment. OT-I (ovalbumin-specific MHC class I-restricted CD8^+^ T cells) mice carrying the transgenic T-cell receptor for SIINFEKL, which is the most immunogenic epitope derived from ovalbumin presented in the context of MHC I, were obtained from Charles River and bred under specific pathogen-free conditions at the VUB. Splenocytes were obtained from OT-I mice between 6 and 10 weeks of age. Ethical approval from the Ethical Committee for Animal Experiments at the VUB was obtained for experiments with CD45.1 mice (21-275-7) and for organ collection of CD45.1 and OT-I mice (20-275-OC1).

### 2.2. Nanoparticle Formulation and Characterization

HRPs LAH4-L1 (KKALLAHALHLLALLALHLAHALKKA) and LAH4-L1-R (RRALLAHALHLLALLALHLAHALRRA) (GenScript, Piscataway, NJ, USA) were purchased as standard phosphate salts. Unmodified and Cyanine5 (Cy5)-labelled methoxyuridine-modified mRNA encoding enhanced green fluorescent protein (eGFP) were purchased from TriLink Biotechnologies (San Diego, USA; L-7601 and L-701, respectively). Unmodified and methyl cytidine (5 mC), pseuoduridine (ψ)-modified mRNA encoding li80tOVA—a truncated form of ovalbumin (tOVA) fused with the first 80 amino acids of the invariant chain (li80) [28]—and Firefly luciferase (Fluc, L-8397, L-8395) were produced by TriLink Biotechnologies. HRPs were mixed with mRNA at 1:1, 1:5, 1:10, and 1:15 ratios (µg mRNA to µg peptide) and allowed to form complexes for 15 min before use in experiments. For LAH4-L1, these ratios were comparable to 0.47:1, 2.36:1, 4.72:1, and 7.08:1 N/P ratios, respectively. For LAH4-L1R, the N/P ratios were 0.45:1, 2.27:1, 4.54:1, and 6.80:1, respectively. The size and ζ-potential of the complexes were determined using ZetaView^®^-NTA (Particle Metrix, Inning am Ammersee, Germany).

### 2.3. Encapsulation Assay

Complexation and protection of mRNA by HRPs was assessed using a gel encapsulation assay. Complexes were made as described above with Fluc mRNA and incubated with 0.4 ng of RNase A (Thermo Fisher Scientific, Brussels, Belgium) per µg of mRNA in 20 mM Tris-HCl, pH 7.6, for 30 min at 37 °C. Subsequently, complexes were treated with 4 µg/µL proteinase K (Merck, Darmstadt, Germany) in 20 mM Tris-HCl in 1% SDS, pH 8, for 30 min at 55 °C, to degrade the HRPs. Intact or interrupted complexes were then loaded on a 1% agarose gel, containing 6% formaldehyde and 0.032 µL per µg agarose Midori Green (Nippon Genetics, Leuven, Belgium) in 3-morpholinopropane-1-sulfonic acid (MOPS) (Sigma-Aldrich, Ghent, Belgium) buffer, separated for 1 h at 50 V and visualized through UV light. The encapsulation efficiency of HRPs was quantified in duplicate using the Quant-it^TM^ RiboGreen RNA assay kit (Thermo Fisher Scientific, Brussels, Belgium). After complexation and treatment with proteinase K, samples were diluted in TAE buffer and mixed volume-to-volume with Quant-it^TM^ RiboGreen working solution. Sample fluorescence was measured with the SpectraMax^®^ iD3 (VWR, Leuven, Belgium), and encapsulation efficiency was determined according to the following formula:$$\mathrm{Encapsulation}\;\mathrm{efficiency}=\frac{\mathrm{mRNA}\;\mathrm{released}-\mathrm{free}\;\mathrm{mRNA}\;\mathrm{after}\;\mathrm{encapsulation}}{\mathrm{uncomplexed}\;\mathrm{mRNA}}\;*\;100\;$$

### 2.4. Cell Culture

The murine dendritic cell line DC2.4 was kindly provided by Prof. P. Midoux from the Centre National de Recherche Scientifique (CNRS) and the Université d’Orléans. DC2.4 cells were cultured in a 37 °C incubator with 5% CO_2_ in Roswell Park Memorial Institute (RPMI) 1640 medium (VWR, Leuven, Belgium) supplemented with 5% fetal bovine serum (FBS) (TicoEurope, Amstelveen, the Netherlands), 50 µM β-mercaptoethanol (Sigma-Aldrich), and a mixture of supplements (SUP) consisting of 50 units/mL and 50 µg/mL penicillin and streptomycin (Sigma-Aldrich), respectively, 2 mM L-glutamine (Sigma-Aldrich), 1 mM sodium pyruvate (Sigma-Aldrich), and 100 µM non-essential amino acid solution (Lonza, Basel, Switzerland), hereafter referred to as complete RPMI.

### 2.5. Bone Marrow Derived Dendritic Cells

Bone marrow was flushed from the femurs and tibias of CD45.1^+^ mice, treated with red blood cell lysis buffer, and washed in Dulbecco’s phosphate-buffered saline (dPBS, VWR). Bone marrow was resuspended at 5 × 10^6^ cells in 10 mL of RPMI containing SUP, 5% fetal clone I serum (FCI) (Thermo Fisher Scientific, Brussels, Belgium), and 50 µM β-mercaptoethanol, hereafter referred to as RPMI+, and supplemented with 20 ng/mL recombinant murine granulocyte–macrophage colony-stimulating factor (rmGM-CSF) (ImmunoTools, Friesoythe, Germany) to generate GM-CSF-derived DCs (GMDCs). GMDCs were cultured in 10 cm Petri dishes at 37 °C and 5% CO_2_. After 3 days, 10 mL of RPMI+ with 40 ng/mL rmGM-CSF was added. Cells were ready for experiments on day 6. CD103^+^ DCs were generated based on the protocol by Tel-Karthaus et al. [29]. Briefly, bone marrow was cultured in Iscove’s Modified Dulbecco’s Medium (IMDM, VWR) containing SUP, 10% FCI, and 50 µM β-mercaptoethanol, and supplemented with 200 ng/mL recombinant human Fms-like tyrosine kinase 3 ligand (rhFlt3L, PeproTech, Hamburg, Germany) and 5 ng/mL rmGM-CSF (PeproTech), hereafter referred to as complete IMDM. CD103^+^ DCs were cultured at 37 °C and 10% CO_2_, at 15 × 10^6^ cells per 10 mL of complete IMDM. On day 3, 5 mL of complete IMDM was added, and on day 6, 5 mL of the cell suspension was refreshed with new complete IMDM and added back to the cell suspension. Non-adherent cells were harvested on day 9 and re-plated in complete IMDM at 5 × 10^6^ in 15 mL in fresh Petri-dishes. The cells were used in experiments on day 13.

### 2.6. Transfection

DC2.4 cells were seeded at a density of 400,000 cells per mL the day before transfection. On the day of transfection, the medium was removed, and the cells were washed once in Opti-MEM^TM^ (Gibco, Thermo Fisher Scientific). Subsequently, complexes, prepared at different ratios of 1:5, 1:10, or 1:15 (µg mRNA:µg peptide), were diluted in Opti-MEM^TM^ before addition to the cells. Opti-MEM^TM^ was removed and exchanged for complete RPMI after 3 to 4 h. DC2.4 cells were transfected in Opti-MEM^TM^ in 24-well plates with 250 ng of mRNA, and harvested after 24 h for experiments to assess different mRNA-to-peptide ratios. Experiments with Cy5-labelled mRNA were performed with 125 ng of mRNA in 48-well plates, and DC2.4 cells were harvested after 4, 24, and 48 h for analysis by flow cytometry. CD103^+^ DCs were plated in Opti-MEM^TM^ in a 24-well plate at 400,000 cells per well. HRP–tOVA mRNA complexes (1:15, unmodified or modified µg mRNA:µg peptide) were diluted in Opti-MEM^TM^ before addition to the cells. After 3 h, Opti-MEM^TM^-containing complexes were removed and replaced with complete IMDM. The next day, transfected cells were harvested and placed in co-culture or analyzed by flow cytometry. As a control for maturation, CD103^+^ DCs were treated with 2.5 µg/mL R848 or 0.5 µg/mL lipopolysaccharide (LPS) (both kindly provided by Prof. Karine Breckpot, VUB, Brussels). As a control for antigen presentation, CD103^+^ DCs were pulsed with 10 µg/mL SIINFEKL peptide (AnaSpec, Fremont, CA, USA) for 2 h and re-plated in fresh complete IMDM. The positive control for transfection was jetMessenger^®^ (Polyplus-transfection^®^, Illkirch, France), used according to the manufacturer’s protocol. Briefly, mRNA was diluted in mRNA buffer before adding the jetMessenger^®^ reagent (µg mRNA: µL jetMessenger^®^, ratio 1:2). JetMessenger^®^–mRNA complexes were immediately added to cells plated in complete RPMI or complete IMDM.

### 2.7. Assessment of Inflammasome Activation

GMDCs were treated with 0.5 µg/mL LPS for 3 h before the addition of 2 mM adenosine 5′-triphosphate (ATP) (Sigma-Aldrich) or HRPs (7, 15 or 30 µg/mL) for 1 h. To inhibit the formation of gasdermin D, cells were treated with 50 µM, 25 µM, 10 µM, 5 µM, or 1 µM NSA (Selleck Chemicals, Houston, Texas, USA) for 30 min before the addition of ATP or HRPs. The supernatant was stored at −20 °C, and IL-1β levels were assessed in the supernatant using the ELISA MAX deluxe kit (BioLegend, San Diego, CA, USA), according to the manufacturer’s instructions. In short, Costar^TM^ high-binding plates (Thermo Fisher) were coated with capture antibody and incubated at 4 °C overnight. The next day, the plates were washed with 0.05% Tween 20 in PBS and blocked in assay diluent A at room temperature. After subsequent washing steps, samples were added and incubated for 2 h at room temperature. Following the addition of the detection antibody for 1 h, streptavidin–HRP solution was added for 30 min. Next, the substrate solution was added for 15 min, and the reaction was stopped using 1N H_2_SO_4_.

### 2.8. In Vitro Proliferation Assay

OT-I splenocytes frozen in CryoStor (STEMCELL technologies, Vancouver, Canada) and stored in liquid nitrogen were thawed in RPMI+ and labelled with 0.5 µM CellTrace™ Violet (CTV, Thermo Fisher Scientific) for 20 min in the dark at 37 °C and 5% CO_2_. Staining was neutralized by adding 5 times the volume of medium and incubating for an additional 5 min. CTV^+^ OT-I splenocytes were subsequently plated in RPMI+ in round-bottomed 96-well plates at 200,000 cells per well. Treated or transfected CD103^+^ DCs were harvested and placed in co-culture at a 1:10 DC:T-cell ratio in a total volume of 200 µL. As a positive control, OT-I splenocytes were stimulated with 2 µg/mL SIINFEKL. After 2 days, 2 µM monensin and 5 µg/mL brefeldin A (BioLegend) were added to the co-cultures to stop cytokine secretion. After 16 h, cells were harvested and analyzed for polyfunctionality, defined as T cells displaying multiple functions on a single cell level by flow cytometry (first described in the context of HIV by Betts et al. [30]). Cells were categorized via Boolean gating containing 0–4 functions: IL-2, TNF-α, and/or IFN-γ production, with or without proliferation (measured as dilution of the CTV signal).

### 2.9. In Vivo Adoptive Transfer Assay

Spleens were harvested from 6–10-week-old female OT-I mice and meshed through a 40 µM nylon filter (pluriSelect, Leipzig, Germany). After two minutes of incubation with red blood cell lysis buffer and washing with dPBS, CD8^+^ T cells were positively selected via magnet-activated cell sorting (MACS) according to the manufacturer’s protocol. Briefly, OT-I splenocytes were resuspended in MACS buffer (0.5% BSA, 2 mM EDTA in PBS), and CD8a microbeads (Miltenyi, Bergisch Gladbach, Germany) were added. After 10 min of incubation at 4 °C, cells were loaded onto an LD column. After flushing the column, CD8a^+^ cells were resuspended in dPBS and stained with 1 µM CTV as described in Section 2.8. Afterwards, CD8a^+^ T cells were diluted in dPBS, and 2 × 10^6^ CD8a^+^ OT-I T cells were injected intravenously into CD45.1^+^ mice. After 2 days, CD45.1^+^ mice were immunized intradermally with 10 µg of modified tOVA mRNA complexed with HRPs or in vivo-jetRNA^®^ (Polyplus-transfection^®^) as a positive control. Uncomplexed mRNA was precipitated and resuspended in 80% Ringer’s lactate as a negative control. HRP–mRNA complexes were diluted in 5% glucose–water, and 50 µL was prepared per mouse. In vivo-jetRNA^®^ was prepared according to the manufacturer’s instructions. Briefly, the mRNA and the reagent were mixed at a 1:1 ratio (µg mRNA: µL in vivo-jetRNA^®^) in RNA buffer, and two injections of 50 µL were prepared per mouse. Three days after immunization, the mice were euthanatized. Inguinal lymph nodes were harvested and incubated with 0.1 U/mL Liberase TL (Roche, Basel, Switzerland) in HBSS (Lonza) on ice. After 30 min at 37 °C, lymph nodes were placed on ice, meshed through a 70 µM nylon filter (pluriSelect), and washed in dPBS before staining for flow cytometry analysis.

### 2.10. Flow Cytometry

The fluorescently labelled antibodies CD11c, CD103, MHC I bound to SIINFEKL (MHCI-S), CD40, CD80, CD86, CD45.2, B220, CD24, and Clec9a were purchased from BioLegend; CD45.1, CD3e, CD8a, IFN-γ, TNF-α, and IL-2 were obtained from BD Biosciences (detailed description in Appendix A). Cells were washed with dPBS and subsequently incubated for 20 min at room temperature in the dark with Fixable Viability Dye eFluor 450 or 780 (1/3000, Thermo Fisher Scientific), diluted in dPBS. Afterwards, various combinations of antibodies diluted in FACS buffer (1% BSA, 0.1% sodium azide in PBS) were added for 30 min at 4 °C in the dark. CD103^+^ DCs were gated based on CD11c^+^, B220^−^, CD103^+^, Clec9a^+^, and CD24^+^ for quality control, and stained for CD40, CD80, and CD86 to assess maturation in combination with CD11c and CD103. An antibody targeting SIINFEKL presented in MHC I (MHCI-S) was used to assess antigen presentation on CD103^+^ DCs. Polyfunctional OT-I CD8^+^ T cells were characterized by CD3^+^, CD8^+^, and CTV^+^, and intracellular cytokine staining was performed for the detection of IL-2, IFN-γ, and TNF-α with the Cyto-Fast^TM^ Fix/Perm buffer set (BioLegend), according to the manufacturer’s instructions. Briefly, cells were fixed for 20 min at room temperature with the Cyto-Fast^TM^ Fix/Perm solution; subsequent washing steps were performed with Cyto-Fast^TM^ Perm/Wash solution. After staining for 30 min at 4 °C, samples were resuspended in FACS buffer. Single-cell lymph node suspensions were gated for CD45.2, CD45.1^+^, CD3^+^, and CD8^+^. Samples were analyzed using the BD LSRFortessa™ (BD Bioscience, Franklin Lakes, NJ, USA).

### 2.11. Data Analysis and Statistics

Flow cytometry data were analyzed using FlowLogic™ (Inivai, Melbourne, Australia). Statistical analysis was performed using the GraphPad Prism software 9.3.1 (GraphPad Software, San Diego, CA, USA).

## 3. Results

### 3.1. Particle Characterization

#### 3.1.1. Encapsulation Efficiency

To assess the capacity for the HRPs to spontaneously form complexes with mRNA, we performed an encapsulation assay. HRPs were complexed to mRNA at various ratios and subsequently treated with RNase A to evaluate the protection through complex formation. Thereafter, proteinase K was added to inactivate RNase A and release the mRNA from the complexes. A 1:1 ratio was considered inadequate for complex formation, since the mRNA was not retained within the gel, and was therefore susceptible to degradation by RNase A. mRNA migration was prevented from a ratio of 1:5 onwards, indicating that stable complexes were formed using LAH4-L1 (Figure 1a) as well as LAH4-L1R (Figure 1b). However, when the complexes were treated with the enzymes, only a faint band was seen for the 1:5 ratio, while clear bands were observed for the 1:10 and 1:15 ratios, indicating that the 1:5 ratio was insufficient to protect the mRNA from degradation. Ratios of 1:10 and 1:15 were therefore considered optimal for both HRPs. In addition, we quantified the encapsulation efficiency, which confirmed the results of the encapsulation assay.

#### 3.1.2. Complexes of HRPs and mRNA form Positively Charged Nanoparticles

In the previous experiment, we excluded the ratio 1:1 since it showed insufficient encapsulation of the mRNA. We continued with the ratios 1:5, 1:10, and 1:15, and assessed the size and charge of the complexes. LAH4-L1–mRNA complexes showed an average size of 225 nm for the 1:15 ratio, which marginally decreased for other ratios. Complexes made with LAH4-L1R reached a size of 195 nm for the 1:15 ratio and fluctuated for other ratios (Figure 2a). Therefore, for the 1:15 ratio there was a trend towards smaller NPs being formed for LAH4-L1R compared to LAH4-L1. For LAH4-L1, we observed a charge of +20.6 mV for the highest ratio and, as expected, the charge declined whenever the amount of peptide added to the complexes was reduced. Remarkably, the charge for the arginine variant was found to be up to +30.7 mV, which is significantly higher than for LAH4-L1, and declined when the amount of peptide added to the complexes was reduced (Figure 2b). In conclusion, although the size of NPs with LAH4-L1 and LAH4-L1R was similar for all of the ratios assessed, LAH4-L1R gave rise to NPs with a higher positive charge across all ratios.

### 3.2. LAH4-L1 Outperforms LAH4-L1R in Transfecting Dendritic Cells

#### 3.2.1. HRP–mRNA Ratio of 1:15 Results in the Most Optimal Transfection Conditions for DC2.4 Cells

Next, we performed transfections in the DC2.4 cell line using eGFP mRNA, to determine which ratio was optimal for further investigation. We previously showed that the 1:5 ratio was not sufficient for efficiently encapsulating mRNA, but as we still observed nanoparticle formation, we assessed whether the NP encapsulation for this ratio would be sufficient for transfection. We did not observe a significant decrease in cellular viability for any of the conditions assessed, indicating low toxicity of the HRPs (Appendix A). As expected, the transfection efficiency for the positive control (jetMessenger^®^) and for the mock conditions (without mRNA) was 89% and 1–2%, respectively (Appendix A). Consistent with the limited mRNA encapsulation efficiency, the 1:5 ratio only resulted in 2–5% transfection efficiency (Figure 3a). For the 1:10 and 1:15 ratios, successful transfection was observed, with 20–30% and 40–60% eGFP-positive cells for LAH4-L1R and LAH4-L1, respectively. Across all ratios, LAH4-L1 clearly outperformed LAH4-L1R, with respect to both the percentage and the median fluorescence intensity (MFI) of eGFP (Figure 3b). Moreover, there was a trend towards a higher MFI for LAH4-L1 at the 1:15 ratio compared to the positive control jetMessenger^®^ (Appendix A). There was no significant difference between the 1:10 and 1:15 ratios in terms of transfection efficiency for both HRPs. However, we did observe a trend towards improved transfection using the 1:15 ratio, especially when evaluating the MFI. Based on these experiments, we decided to proceed with the 1:15 ratio.

#### 3.2.2. Similar Uptake but More Efficient Translation for LAH4-L1-Formulated mRNA in DC2.4 Cells

To simultaneously assess the uptake and translation of mRNA, DC2.4 cells were transfected with Cy5-labelled mRNA encoding eGFP. Already, 4 h after transfection, 100% of the complexes were taken up, and remained detectable for up to 48 h (Appendix A). However, by then, the intensity of the Cy5 signal had decreased (Appendix A). For both HRPs, eGFP/Cy5 double-positive cells were already detected after 4 h, indicating that a significant part of the mRNA that is taken up is translated (Figure 3c). mRNA translation increased for both mRNA–HRP NPs after 24 h, remaining stable for up to 48 h. However, the Cy5 signal persisted, indicating that not all mRNA is translated after uptake. Based on the MFI of eGFP, we noticed that transfection with LAH4-L1 already resulted in high translation levels after 4 h (Figure 3d). In contrast, LAH4-L1R only showed similar translational levels after 24 h. Overall, across all time points, LAH4-L1 showed better transfection and translation characteristics than LAH4-L1R in DC2.4 cells.

#### 3.2.3. LAH4-L1 Shows Better Transfection Efficiency Than LAH4-L1R in Primary CD103^+^ DCs

In order to validate our results in a more relevant ex vivo model, we generated CD103^+^ DCs from murine bone marrow and performed a transfection with mRNA encoding tOVA (gating strategy in Appendix A). In addition, modified mRNA was used to further increase the translation efficiency. CD103^+^ DCs pulsed with SIINFEKL were used as positive controls and, as expected, the cells efficiently presented this peptide in the context of MHC I. Upon transfection with jetMessenger^®^—a transfection agent that previously showed efficient transfection with mRNA encoding eGFP in DC2.4 cells (Appendix A)—SIINFEKL was not efficiently presented in MHC I, even when using modified mRNA. As expected, LAH4-L1 complexed to modified mRNA performed significantly better than LAH4-L1 or LAH4-L1R NPs containing unmodified mRNA (Figure 4a). Surprisingly, there was no significant difference in antigen presentation between LAH4-L1 and LAH4-L1R complexed with modified mRNA. For jetMessenger^®^ as well as for the HRPs, the MFI values for presentation of SIINFEKL after transfection with modified mRNA tended to be higher than for unmodified mRNA (Figure 4b). As was observed for DC2.4 cells, LAH4-L1 outperformed LAH4-L1R in transfecting CD103^+^ DCs, but this difference was less pronounced.

### 3.3. HRPs Prepare DCs to Induce an Efficient CD8^+^ T-Cell Response

#### 3.3.1. HRPs Induce Maturation of CD103^+^ DCs

In addition to antigen presentation, co-stimulatory molecules are also important determinants of the efficiency of T-cell stimulation. Therefore, we assessed the expression of the co-stimulatory molecules CD40 and CD86 on the cell surface of CD11c^+^ CD103^+^ DCs after transfection. We used R848 (a Toll-like receptor (TLR)7 agonist) and LPS (a TLR4 agonist) as positive controls for upregulation of CD40 and CD86. Treatment with both of these agents resulted in efficient maturation of CD103^+^ DCs, as indicated by a significant increase in the percentages of CD40 and CD86 compared to the untreated condition. All of the transfection conditions where unmodified mRNA was used showed a significantly increased expression of CD40 and CD86 (Figure 5a). For jetMessenger^®^, no maturation was observed after transfection with modified mRNA, confirming that unmodified mRNA can induce maturation of DCs, and that this maturation effect is abrogated by using modified mRNA, as has been extensively described [31,32]. However, remarkably and unexpectedly, HRPs were able to induce the expression of co-stimulatory molecules in CD103^+^ DCs, irrespective of whether modified or unmodified mRNA was used. Even without mRNA (the mock condition), we observed similar levels of maturation (Figure 5b), indicating that the HRPs themselves could be responsible for this maturation phenomenon.

#### 3.3.2. HRPs Are Capable of Inducing IL-1β Secretion in GMDCs

Previous work has shown that the antimicrobial peptide LL-37/cathelicidin activates the inflammasome pathway [21,33]. LL-37 and HRPs are similar to the extent that they are able to interfere with microbial membranes inducing lysis of bacteria. To assess the ability of the HRPs to activate the inflammasome, we first primed GMDCs with LPS, which is a potent inducer of the transcription of pro-IL-1β (signal 1). LPS treatment followed by ATP (signal 2) led to the release of very large amounts of IL-1β, as expected. We next assessed the ability of the HRPs to induce the assembly of the inflammasome. Incubating GMDCs with the HRPs alone did not result in the secretion of IL-1β. However, when the cells were first primed with LPS, large amounts of IL-1β were released (Figure 6a). In addition, we observed a dose-dependent relationship, where increasing the amount of HRPs led to increased IL-1β secretion. Interestingly, LAH4-L1 and LAH4-L1R induced a similar secretion of the pro-inflammatory cytokine. To confirm that IL-1β release was related to the activation of the inflammasome, GMDCs were treated with increasing concentrations of NSA—a known blocker of gasdermin D—in between LPS stimulation and HRP incubation. For both HRPs, we observed a dose-dependent inhibition of IL-β secretion by NSA (Figure 6b), indicating that blocking gasdermin D pore assembly downstream of inflammasome activation inhibits the release of IL-1β. Overall, we can conclude that HRPs are capable of initiating inflammasome activation, provided that the required components are present in the cells.

### 3.4. HRPs Induce a CD8^+^ T-Cell Response In Vitro and In Vivo

#### 3.4.1. HRP-Transfected DCs Induce Polyfunctionality in CD8^+^ T Cells

We observed that HRPs induce maturation in DCs, which is an important prerequisite for T-cell activation. Next, we assessed the ability of the transfected CD103^+^ DCs to stimulate T cells by co-culturing them with OT-I splenocytes. Mock-transfected or untreated DCs did not induce proliferation of OT-I cells; therefore, polyfunctionality was not assessed for these conditions (Appendix A). OT-I splenocytes placed in co-culture with transfected CD103^+^ DCs all displayed proliferation (defined as T cells having divided more than three times), as well as IL-2, TNF-α, and IFN-γ production, referred to as polyfunctionality (Appendix A). Consistent with the low levels of antigen presentation in combination with low levels of maturation observed for the DCs transfected with jetMessenger^®^, we also observed low levels of T-cell polyfunctionality within the CD8^+^ T-cell subset in the co-cultures (Figure 7a). In addition, the conditions with the HRPs all showed higher levels of polyfunctionality compared to the jetMessenger^®^ conditions. Remarkably, we observed a significant difference only between LAH4-L1 and LAH4-L1R complexed to modified mRNA compared to jetMessenger^®^ complexed to unmodified mRNA. However, little difference was noted between LAH4-L1 and LAH4-L1R, or between unmodified and modified mRNA. Although modified mRNA clearly increased antigen presentation levels, and the HRPs induced sufficient upregulation of co-stimulatory molecules, polyfunctionality levels for T cells in the co-cultures were equally high for both unmodified and modified mRNA. For the production of IL-2 in CD8^+^ T cells, we noticed the same trend as for the polyfunctionality. The HRP-transfected CD103^+^ DCs induced more IL-2 production than jetMessenger^®^-transfected DCs, but no differences could be observed between LAH4-L1 and LAH4-L1R (Figure 7b). We summarized the data for polyfunctionality, ranging from 0 to 4 functions—namely, proliferation, and production of TNF-α, IFN-γ, and/or IL-2—on a single-cell level, using a pie chart for each of the conditions. The positive controls, CD103^+^ DCs pulsed in co-culture with OT-I splenocytes, or OT-I splenocytes directly pulsed with SIINFEKL, resulted in efficient induction of polyfunctional CD8^+^ T cells (Appendix A). Pie charts displaying the presence of 0–4 functions reveal that there are fewer CD8^+^ T cells displaying no functions when co-cultured with CD103^+^ DCs transfected with HRPs and modified mRNA compared to unmodified mRNA (Figure 7c). In addition, the proportion of CD8^+^ T cells exhibiting more than two functions showed a trend towards being larger for conditions using modified mRNA. In conclusion, the optimal co-stimulation and antigen presentation by CD103^+^ DCs transfected with modified mRNA complexed to LAH4-L1 or LAH4-L1R resulted in similar levels of polyfunctional CD8^+^ T cells.

#### 3.4.2. HRP–mRNA Complexes Induce Proliferation of OT-I CD8^+^ T Cells In Vivo

In the next step, we assessed the HRP–mRNA complexes in vivo. Two days after the adoptive transfer of CTV-labelled CD45.2^+^ CD8^+^ OT-I cells into CD45.1^+^ mice, the CD45.1^+^ mice were injected intradermally with the mRNA–HRP NPs, uncomplexed mRNA, or mRNA complexed to in vivo-jetRNA^®^, previously used as an immunization reagent [34]. Three days later, the proliferation of transferred OT-I splenocytes was assessed. We used the CD45.2 and CD45.1 mismatch to efficiently trace back the CD45.2^+^ OT-I splenocytes (Figure 8a). In vivo-jetRNA^®^ led to solid proliferation of CD8^+^ OT-I splenocytes. Although both HRPs performed better than RNA alone, they did not outperform in vivo-jetRNA^®^ in an in vivo setting (Figure 8b). Even though the difference was not statistically different, the arginine variant seemed to outperform the lysine variant. In conclusion, the arginine variant took the upper hand and performed similarly to our positive control, reversing the roles of the HRPs compared to what we observed in vitro.

## 4. Discussion

mRNA has been shown to be a promising tool for vaccination, as demonstrated by the recent success of two COVID-19 vaccines [35,36]. The mechanism responsible for the adjuvanticity of these mRNA vaccines, however, is only beginning to be unraveled. In addition, in general, because arginine residues occur in natural nucleic-acid-binding motifs, they have been assumed to perform better in the formation of NPs than lysine residues [13,15,37,38]. However, the transfection efficiency of arginine- or lysine rich peptide mRNA complexes has not been directly compared. Here, we addressed the question whether HRPs based on lysine or arginine residues performed better in terms of mRNA NP formation, transfection efficiency, and immunogenicity.

Firstly, we determined the optimal mRNA/peptide ratio needed to protect mRNA in the presence of RNases. We noticed that for both peptides, complexes were formed from a ratio of 1:5 onwards. However, for protection from RNases, mRNA/peptide ratios from 1:10 onwards were needed. Although we noticed that the ability to encapsulate mRNA was the same for LAH4-L1 or LAH4-L1R, the NPs formed by these two HRPs showed different characteristics. The particles formed with LAH4-L1–mRNA complexes had a lower ζ-potential than the LAH4-L1R–mRNA complexes. The latter can possibly be explained by the fact that the arginine-based peptide LAH4-L1R packages the mRNA more densely than LAH4-L1, as previously shown for lysine and arginine polypeptides packaging DNA [39]. In this way, LAH4-L1R results in more compact mRNA–NPs with a higher positive charge. Furthermore, the tighter grip of the arginine residues on the mRNA could also contribute to the lower transfection efficiency, as the mRNA could be less readily available for translation in the cytoplasm. By using Cy5-labelled mRNA, we simultaneously investigated the uptake and translation of the mRNA. For both HRPs, the uptake of NPs was 100%; however, not all mRNA was translated. These results are comparable with those of other mRNA NP formulations—for example, those consisting of DOTAP/DOPE [40]. One of the reasons for this might be that by releasing the mRNA in the cytoplasm, it becomes susceptible to endogenous RNases. It is possible that these RNases make small cuts in the mRNA, retaining the Cy5 signal but rendering the mRNA unable to be translated. In this case, the tighter grip of arginine residues on the mRNA could be beneficial, leading to better protection of the mRNA from RNases, as was shown for disulfide crosslinking of mRNA in polyplex micelles [41]. However, it is more likely that the mRNA is stuck in the endosome, as was shown for lipid nanoparticles (LNPs). After uptake, the mRNA-LNPs were targeted towards different endosomal compartments. Therefore, some of the LNPs ended up in the early/recycling endosomes, promoting endosomal escape of the mRNA molecules. The late endosomes, lysosomes, or arrested endosomes, on the other hand, are counterproductive for endosomal escape, and are the cause of the remaining mRNA in the cell [42]. Nevertheless, we observed that the intensity of the eGFP signal per cell was already high 4 h after transfection with LAH4-L1–mRNA complexes. Although initially the intensity lagged behind for the LAH4-L1R–mRNA NPs, it caught up after 24 h, further supporting the hypothesis that arginine residues lead to a less efficient release of mRNA compared to lysine residues. Afterwards, the MFI for eGFP remained high for both peptides, slowly decreasing after 48 h, which was expected due to the half-life of eGFP, which is approximately 24 h [43]. Because the percentage of eGFP was higher for LAH4-L1, we still concluded that the latter peptide was better for transfecting DC2.4 cells

To assess the HRP–mRNA NPs in a more relevant context, we generated CD103^+^ DCs ex vivo. These DCs resemble the migratory cross-presenting DC population and, therefore, act as a better model than the GMDCs. In fact, the GMDCs have a monocyte-derived inflammatory DC phenotype, which is less relevant in the setting of antigen presentation and T-cell activation. Even though we transfected the DCs with low doses of mRNA, efficient antigen presentation was still detected—especially in case of the HRP–mRNA NPs. To further increase the translational potential of the HRP–mRNA NPs, we used modified mRNA, which has been shown to result in more efficient translation, but as a side effect was also less immunogenic [31,32,44]. Indeed, the antigen presentation increased when using modified mRNA—again, mainly in HRP–mRNA nanoparticle-transfected cells. Using jetMessenger^®^—the positive control—the immune-stimulatory capacity of unmodified mRNA was undeniable, leading to an upregulation of CD40 and CD86. However, for the HRPs, the maturation effect could not solely be attributed to the mRNA since HRPs alone also efficiently induced upregulation of CD40 and CD86. We hypothesize that this was due to ion fluxes in the cell after disruption of the cellular membranes. After endocytosis, the HRPs are influenced by the acidification of the environment to adopt a membrane in planar formation [45]. Thus, the cell-penetrating capacity of HRPs is able to create holes in the endolysosome, primarily to release mRNA cargo. In this way, pore formation by toxins, for example—and in this case by HRPs—can lead to K^+^ efflux not only from the lysosome, but also through pores and channels in the plasma membrane [46]. This leakage and K^+^ release from the cytosol has been shown to upregulate maturation markers in DCs for toxins [47].

The third signal in the DC–T-cell interaction required for efficient activation of the immune response is delivered by cytokines. Nanoparticles consisting of LAH4-L1–mRNA complexes combined with PLA were shown to induce IL-1α, IL-6, IFN-α, TNF-α, and IL-12p70 in monocyte-derived DCs, indicating a pro-inflammatory Th1 profile [16]. However, it is unclear whether this effect was due to the LAH4-L1 component itself or to PLA. Moreover, the pore formation of LAH4-L1 may induce leakage of cathepsins from the lysosome, resulting in the formation of reactive oxygen species—another danger signal for the cells [48]. In combination with the K^+^ efflux described above, these pathways converge in the oligomerization of inactive NLRP3 by recruiting NIMA-related kinase 7 (NEK7), apoptosis-associated speck-like protein containing a caspase activation and recruitment domain (CARD) (ASC), and caspase-1, resulting in the cleavage of pro-IL-1β, pro-IL-18, and gasdermin D into their active forms. Therefore, similar to what has been observed for LL-37 in previous studies, we found that HRPs are also very efficient in activating the inflammasome, resulting in the release of IL-1β, after priming with LPS. Although we did not specifically evaluate which inflammasome was activated in this paper, the combination of signals proposed strongly indicates the involvement of the NLRP3 inflammasome, and not NLR family CARD domain containing 4 (NLRC4) or absent in melanoma 2 (AIM2).

Inflammasome activation always requires a priming and activation stimulus. Here, we provided priming with LPS and activation with the HRPs. However, we and others hypothesize that mRNA is able to deliver the first signal. mRNA–DOTAP/DOPE NPs have already been shown to induce the transcription of IL-1β and IL-6 (an IL-1β-instructed gene) in bone-marrow-derived DCs [49]. In this case, we could argue that the use of unmodified mRNA is pro-inflammatory, leading to induction of these cytokines. However, N1-methylpseudouridine (N1mψ)-modified mRNA packaged in a cationic polymer–liposome hybrid nanoparticle induced transcription of IL-1β and IL-6 in the blood of mice without a strong type-I IFN response [50]. As for the second signal, LNPs designed in a similar fashion as for the COVID-19 mRNA vaccines—but without mRNA—induced a strong pro-inflammatory cytokine profile (including IL-1β and IL-6) in mice upon injection into the skin [51]. Indeed, mRNA–liposome NPs induced IL-1β secretion in human peripheral blood mononuclear cells, induced by activation of the NLRP3 inflammasome [52]. However, the inclusion of N1mψ-modified mRNA abrogated this induction. In contrast, the COVID-19 vaccines containing N1mψ-modified mRNA, formulated in LNPs containing either MC3 (BioNTech) or SM-102 (Moderna), were able to induce IL-1β in human PBMCs. This indicates that signal 1 for mRNA is a complicated case, and depends on the length as well as on several secondary structures in the mRNA that could stimulate the transcription of genes of pro-inflammatory cytokines, most likely due to the activation of melanoma differentiation-associated protein 5 (MDA5) [53]. In addition, inflammasome activation and immune stimulation present a delicate balance. Low IL-1β is more beneficial in the immunological synapse, while a strong IL-1β response is associated with pyroptosis and less efficient CD8^+^ T-cell responses [54]. By pushing the cells too much towards inflammasome activation, massive pyroptosis may occur. This could be the reason why combining liposomes and LAH4-L1 for the formulation of mRNA resulted in increased cellular toxicity compared to NPs with either LAH4-L1 or liposome alone [55]. Keeping this in mind, it is important to note that the immune synapse formed between T cells and DCs already brings these cells in close proximity, indicating that low amounts of IL-1β secreted by DCs might be sufficient to activate T cells [56].

The DC maturation induced by the HRPs, in combination with the efficient antigen presentation, provides a good basis for T-cell stimulation. In the case of jetMessenger^®^–mRNA complexes, too little antigen presentation with unmodified mRNA resulted in very low induction of T-cell polyfunctionality in the co-culture assay. Complexes of jetMessenger^®^ with modified mRNA showed slightly better antigen presentation levels, but no upregulation of co-stimulatory molecules. However, the low baseline presence of co-stimulatory molecules was sufficient to increase the levels of polyfunctionality. For the HRPs, even though NPs packaging unmodified mRNA resulted in low antigen presentation, the co-stimulation still led to a high induction of proliferation in T cells, in combination with cytokine production. When combining the HRPs with modified mRNA, the polyfunctionality was boosted further by improving the antigen presentation capacity. IL-2 production was remarkably high in CD8^+^ T cells after co-culture with HRP–mRNA-transfected DCs. This is especially interesting in the context of both therapeutic HIV vaccination and tumor immunology, where a reduction in IL-2 production is one of the first manifestations of exhaustion in CD8^+^ T cells [57,58].

In vivo results indicated that LAH4-L1R outperformed LAH4-L1 in the induction of proliferation of CD8^+^ T cells. As discussed above, we hypothesize that the tighter grip of LAH4-L1R on the mRNA results in better protection from RNases—in this case, in an in vivo setting—resulting in increased antigen presentation and improved T-cell activation. However, it has already been shown that LAH4-L1 and LAH6-L1-80 perform better in DNA transfection in the presence of serum than other LAH4 variants [59]. Although LAH4-L1R was not assessed, this still indicates that LAH4-L1 has some serum stability capacities. Another possibility is that the higher availability of arginine in the lymph nodes, after migration of the NPs, results in a metabolic shift in the T cells, instructing them to survive and proliferate [60]. To further improve the HRP–mRNA NPs in an in vivo setting, we could opt for PEGylation or acylation of the HRPs, or complexation with other NPs such as LNPs or polyelectrolyte particles [7].

In conclusion, we showed that LAH4-L1 outperforms LAH4-L1R in transfecting the DC2.4 cell line as well as primary CD103^+^ DCs, in contrast with the hypothesis that arginine-based peptides would outperform lysine-based peptides in terms of nucleic acid delivery. In addition to the priming signals from mRNA, the HRPs were able to induce the second signal resulting in activation of the inflammasome, which could provide intrinsic adjuvant activity for vaccines.

## Figures and Tables

**Figure 1 pharmaceutics-14-01387-f001:**
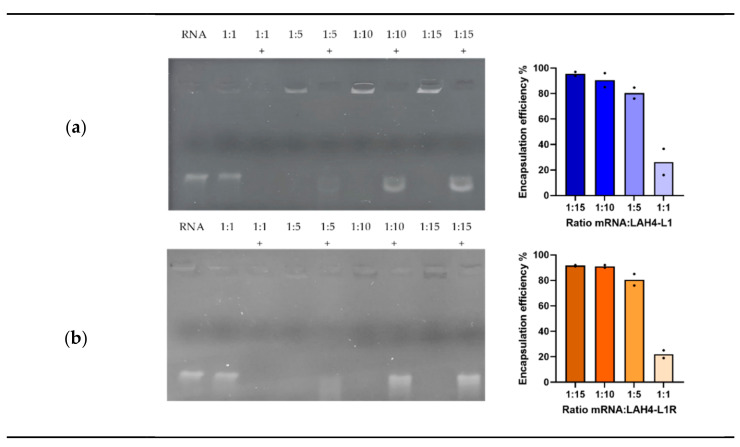
Encapsulation efficiency of histidine-rich peptides: mRNA was complexed to LAH4-L1 (**a**) or LAH4-L1R (**b**) at the indicated ratios (µg mRNA: µg peptide) and subjected to treatment with RNase A and proteinase K (indicated with a “+”). mRNA released from the complexes was visualized by gel electrophoresis (representative result, *n* = 3). The percentage encapsulation efficiency was determined based on the amount of free mRNA after complexation and the mRNA released from the complexes after treatment with proteinase K. mRNA concentration was determined with the Quant-it^TM^ RiboGreen assay kit (*n* = 2, mean, each dot represents one experiment).

**Figure 2 pharmaceutics-14-01387-f002:**
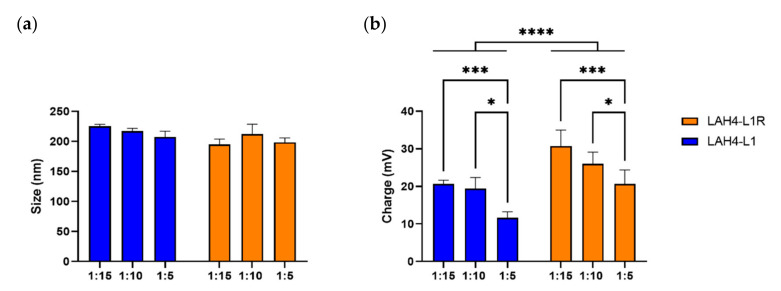
Characterization of histidine-rich peptide (HRP)–mRNA nanoparticles. Different ratios of mRNA:HRP (µg:µg) were prepared for LAH4-L1 and LAH4-L1R. Size (**a**) and charge (**b**) were determined via ZetaView^®^–NTA (*n* = 3, mean ± SD). Two-way ANOVA, Tukey’s multiple comparison test: * *p* < 0.05, *** *p* < 0.001, **** *p* < 0.0001.

**Figure 3 pharmaceutics-14-01387-f003:**
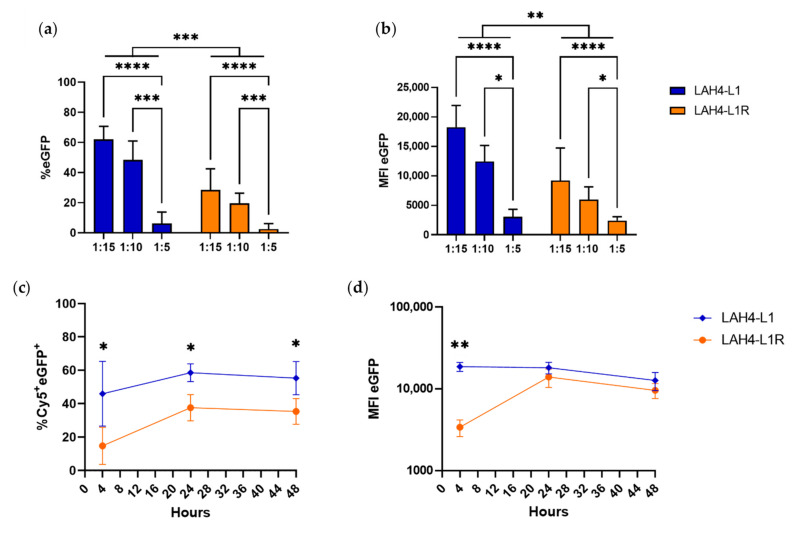
Histidine-rich peptides allow efficient transfection in DC2.4 cells. DC2.4 cells were transfected with 250 ng of mRNA encoding eGFP per 200,000 cells at the indicated mRNA:HPR ratios (µg:µg). The next day, the cells were harvested, and readout was performed by flow cytometry for percentage (**a**) and MFI (**b**) of eGFP (*n* = 4, mean ± SD). Two-way ANOVA, Tukey’s multiple comparison test: * *p* < 0.05, ** *p* < 0.01, *** *p* < 0.001, **** *p* < 0.0001. DC2.4 cells were transfected at a 1:15 ratio (µg mRNA:µg HRP) with 125 ng of Cy5-labelled mRNA encoding eGFP per 100,000 cells. Readout was performed after 4, 24, and 48 h by flow cytometry for detection of the percentage of Cy5^+^eGFP^+^ cells (**c**) and the MFI of eGFP (**d**) (*n* = 3, mean ± SD). Welch’s t-test: * *p* < 0.05, *** *p* < 0.001. The positive control (jetMessenger^®^) and negative controls (without mRNA) are not shown on the graph.

**Figure 4 pharmaceutics-14-01387-f004:**
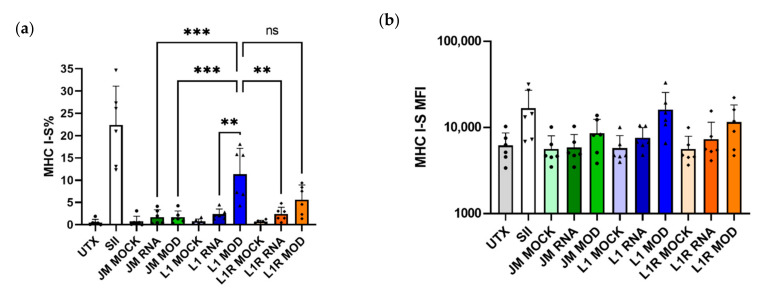
Antigen presentation of bone-marrow-derived CD103^+^ DCs after transfection with LAH4-L1 (L1)- or LAH4-L1R (L1R)-formulated tOVA mRNA. CD103^+^ DCs were generated and transfected on day 13 with mRNA encoding tOVA (unmodified, RNA/modified, MOD). The next day, cells were harvested and stained for DC markers and SIINFEKL presented in MHC I (MHCI-S). Percentage (**a**) and MFI (**b**) for SIINFEKL presented in MHC I within the CD11c^+^ CD103^+^ population; cells were pulsed with SIINFEKL peptide (SII) as a positive control; positive control for transfection was performed with jetMessenger^®^ (JM). Mock controls were performed without mRNA. *n* = 6; each dot represents one replicate, mean ± SD. One-way ANOVA, Tukey’s multiple comparison test: ** *p* < 0.01, *** *p* < 0.001, ns—not significant.

**Figure 5 pharmaceutics-14-01387-f005:**
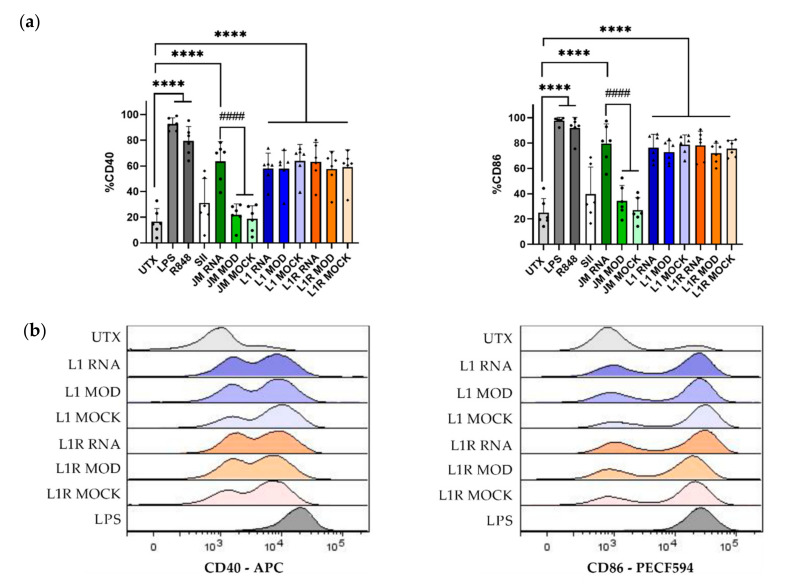
Maturation of bone-marrow-derived CD103^+^ DCs after transfection with LAH4-L1 (L1)- or LAH4-L1R (L1R)-formulated tOVA mRNA. CD103^+^ DCs were left untreated (UTX) or transfected on day 13 with mRNA encoding tOVA (unmodified, RNA/modified, MOD), or without mRNA (MOCK). The next day, cells were harvested and stained for DC markers and the maturation markers CD86 and CD40. Percentage CD40 and CD86 expression within the CD11c^+^ CD103^+^ DCs (**a**); cells were treated with LPS or R848 as a positive control; positive control for transfection was performed with jetMessenger^®^ (JM); positive control for antigen presentation where DCs were pulsed with SIINFEKL (SII). Overlay graphs for CD40 and CD86 expression (**b**); one representative result. *n* = 6; each dot represents one replicate, mean ± SD. One-way ANOVA, Tukey’s multiple comparison test: **** *p* < 0.0001 compared to the untreated cells; #### *p* < 0.0001 compared to jetMessenger^®^-mRNA.

**Figure 6 pharmaceutics-14-01387-f006:**
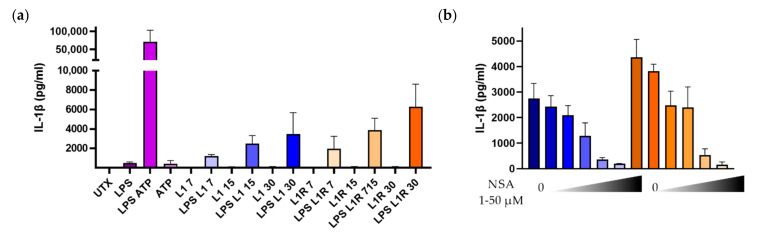
Inflammasome activation in GMDCs treated with LPS and histidine-rich peptides. GMDCs were generated by culture with 20 ng/mL GM-CSF for 6 days. On day 6 (**a**), GMDCs were left untreated (UTX) or stimulated with 0.5 µg/mL LPS for 3 h, followed by 1 h of incubation with 2 mM ATP, or with LAH4-L1 (L1) or LAH4-L1R (L1R) at various concentrations (7, 15, or 30 µg/mL). The supernatant was harvested, stored at −20 °C, and analyzed via an ELISA for IL-1β. After LPS stimulation, GMDCs were incubated with 0–50 µM NSA before treatment with 15 µg/mL HRPs for 1 h (**b**). (*n* = 3), mean ± SD.

**Figure 7 pharmaceutics-14-01387-f007:**
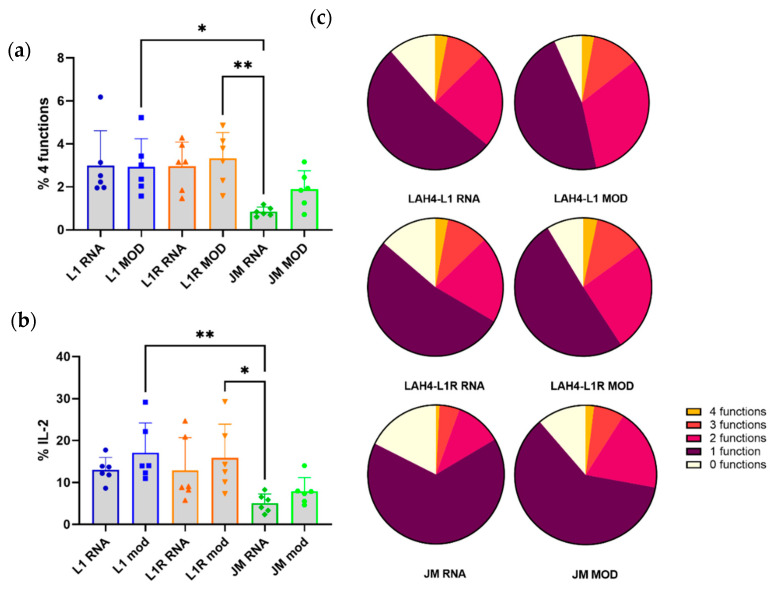
Transfected CD103^+^ DCs induce polyfunctionality in OT-I splenocytes ex vivo. Transfected CD103^+^ DCs were co-cultured with OT-I splenocytes for 3 days. On the second day, monensin and brefeldin A were added overnight to block cytokine secretion. The next day, polyfunctionality was assessed within the CD8^+^ T-cell population. The 4 functions assessed were proliferation (dilution of CTV) and TNF-α, IFN-γ, and IL-2 production. Negative controls included co-culture with untreated or mock-treated DCs, and positive controls comprised SIINFEKL-pulsed DCs or SIINFEKL directly added to the splenocytes (conditions not shown). CD8^+^ T cells were assessed for the co-expression of the 4 functions (**a**) and the percentage of CD8^+^ T cells producing IL-2 (**b**). Polyfunctionality was summed up in pie charts, with each part of the whole representing the mean proportion of cells expressing 0–4 functions (**c**). *n* = 6; each dot represents one replicate, mean ± SD. One-way ANOVA, Tukey’s multiple comparison test: * *p* < 0.05, ** *p* < 0.01.

**Figure 8 pharmaceutics-14-01387-f008:**
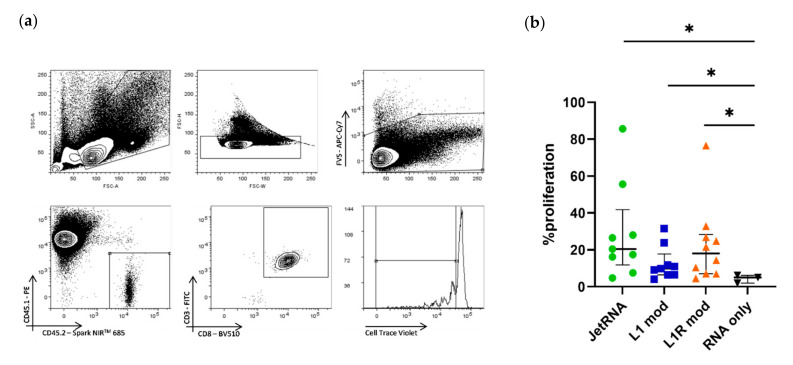
Immunization with histidine-rich peptide–mRNA formulations results in proliferation of CD8^+^ OT-I T cells. CD45.1 mice were injected with CTV-labelled CD45.2^+^ CD8^+^ OT-I splenocytes, and 2 days later they were immunized with 10 µg of modified mRNA encoding tOVA complexed to either in vivo-jetRNA^®^ (JetRNA), LAH4-L1 (L1 mod), or LAH4-L1R (L1R mod). After 3 days, lymph nodes were harvested, and single-cell suspensions were analyzed by flow cytometry. Gating was performed on viable cells; subsequently, we looked for CTV dilution in the CD45.2^+^ CD3^+^ CD8^+^ T-cell population (**a**). Percentage of CD3^+^ CD8^+^ T-cell proliferation (**b**). Each symbol represents one mouse; median +/− IQR. Mann–Whitney test: * *p* < 0.05.

## Data Availability

The raw data supporting the conclusions of this article will be made available by the authors upon request.

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
