# Peer review of "Efficient Induction of Antigen-Specific CD8^+^ T-Cell Responses by Cationic Peptide-Based mRNA Nanoparticles"

_pharmaceutics, 2022, doi:10.3390/pharmaceutics14071387_

Round 1
Reviewer 1 Report
This article explores the differences in the effects of mRNA vaccine delivery peptide LAH4-L1 and its arginine variant LAH4-L1R in mRNA delivery, and validates their potential application in immune adjuvants, which provides valuable experience for the design of mRNA delivery polypeptides. However, there are still some small problems in the article that need to be corrected:
1. The introduction part describes a lot of irrelevant content, so that the main line is not prominent enough, such as “In previous research, it has already been shown that LAH4-L1 is an efficient delivery tool for small interfering (siRNA) and pDNA in vitro and in vivo [15–17]…Interestingly, LAH4 complexing OVA peptides in combination with the adjuvant CpG was able to induce a stronger CD8+T cell response than OVA-CpG alone [18].” and “A well-known naturally occurring member of this class of peptides is LL-37 or cathelicidin…pro-IL-18 and pro-IL-1β, giving rise to secretion of inflammatory cytokines as well as pyroptosis [21]”. It is suggested to reduce the narrative of previous literature to highlight the main line of the article.
2. The characterization of LAH4-L1-mRNA and LAH4-L1R-mRNA is crude. It is recommended to supplement at least transmission electron microscopy images of nanoparticles to illustrate the formation of nanoparticles.
3. There are problems in the statistical tests of some figures in the article, such as Figure 2b, Figure 3b, Figure 5a, etc. Some of these difference comparisons seem to be problematic. A recalculation confirmation is recommended.
4. The mRNA delivery strategy mentioned in the article (Several formulation strategies have been investigated for efficient delivery of nucleic acids, including polymers [4], cationic peptides [5] and lipid formulations [6].) is not complete. Other mRNA delivery strategies include delivery methods based on metal ions and inorganic nanoparticles. It is recommended to supplement relevant literature, such as:
(1). doi: 10.1016/j.cej.2020.126080
(2). doi: 10.1186/s12951-021-01212-9
5. In the discussion section it is mentioned that "The particles formed with LAH4-L1 had a lower ζ-potential. The latter can possibly be explained by the fact that the arginine-based peptide LAH4-L1R packages the mRNA more densely than LAH4-L1, as previously shown for DNA [38].", but whether the difference in zeta potential is caused by the difference in the peptide itself. It is suggested to supplement the zeta potential of individual polypeptides to illustrate this problem.
Reviewer 2 Report
Joeri L. Aerts and coworkers described an efficient carrier system to deliver mRNA into dendritic cells. The authors compared mRNA polyplexes formulated with histidine-rich peptides containing lysine and arginine, LAH4-L1 (KKALLAHALHLLALLALHLAHALKKA) and LAH4-L1-R (RRALLAHALHLLALLALHLAHALRRA), respectively, for efficient translation and induction of antigen-specific CD8+ T cell responses.
Indeed, with a battery of well-done experiments, the authors demonstrated the advantage of LAH4-L1 over LAH4-L1-R in transfecting DC2.4 and primary CD103+ DCs.
Altogether, the authors provide an excellent study worth publishing in MDPI Pharmaceutics. However, I recommend a detailed revision addressing the following aspects carefully:
1. Many acronyms were used without expansion. Please expand the acronym after its first appearance in the text.
OT-I OT-I (ovalbumin-specific MHC class I-restricted CD8+ T cells)
li80tOVA a truncated form of ovalbumin (tOVA) fused with the first 80 amino acids of the invariant chain (Ii80)
NLRP3 Nucleotide-binding and oligomerization domain-like receptor family pyrin domain containing 3
eGFP enhanced green fluorescent protein
There is no need for the “siRNA” acronym as small interfering was mentioned only once in the text. Similar to the case for necrosulfonamide (NSA). Too many acronyms in the manuscript also divert the attention of readers.
2. poly-L-Arginine and poly-L-lysine The prefix levorotatory isomer should be in small font size than the usual font. “poly-L-Arginine and poly-L-lysine.”
3. In the abstract authors used the following sentence: "Cationic peptides present interesting candidate NPs, ..." .
Cationic peptides are not nanoparticles. Instead, they are carriers for mRNA delivery. Please change the sentence.
4. In the Reference section, the authors used MRNA. Please use mRNA.
5. How is the mRNA released from complexes of LAH4-L1 and LAH4-L1-R? Is proteinase K treatment digest the LAH4-L1 and LAH4-L1-R catiomers facilitated the release of mRNA?
6. Is there any difference in the serum tolerability of mRNA complexed with LAH4-L1 and LAH4-L1-R? Quantitative PCR can answer this. The authors used RNase A tolerability using gel electrophoresis. I am not suggesting to provide this data in the current manuscript. However, I request the authors to keep this in mind for future works.
7. Define clearly the word polyfunctionality after its first appearance. Please mention the functions in “0 to 4 functions” right after its first appearance in the manuscript. Polyfunctionality is the ability of T cells to carry out multiple functions, such as two or more cytokines (IFN-γ, TNF-α, IL-2 simultaneously) or degranulation of cytotoxic proteins simultaneously at the single-cell level.
8. The authors mentioned the complexation ratio in terms of µg:µg. Please also note in the conventional way that is Nitrogen to Phosphate (N/P) ratio.
9. The authors discussed that “LAH4-L1R results in more compact mRNA-NPs with a higher positive charge”. But the size data show no apparent difference between LAH4-L1R and LAH4-L1.
10. The particles formed with LAH4-L1 had a lower ζ-potential. The latter can possibly be explained by the fact that the arginine-based peptide LAH4-L1R packages the mRNA more densely than LAH4-L1, as previously shown for DNA. In the above sentences, what is the latter?Probably the discussion should be as shown below. The particles formed with LAH4-L1 had a low ζ-potential. In marked contrast, the particles of LAH4-L1R presented high ζ-potential. The latter can possibly be explained by the fact that the arginine-based peptide LAH4-L1R packages the mRNA more densely than LAH4-L1, as previously shown for DNA.
Here, the latter is about the LAH4-L1R complex.
11. Sometimes mentioned that data is not shown. I welcome the authors to show such vital data in Supplementary information.
12. Please provide the citation for “For both applications, it is crucial that the formulation increases cellular uptake and ensures endosomal escape.” One suggestion is a recent article addressing the endosomal escape of mRNA polyplex 10.1002/marc.202100754.
13. The authors mentioned that the persisted Cy5 signal indicates that not all mRNA is translated after uptake. Give the reason for this critical observation. One plausible reason could be the susceptibility of mRNA cleavage by intracellular ribonucleases; even a single cut in the reading frame of mRNA by RNases would end up with no protein production. The authors discussed that the tighter grip of the arginine residues on the mRNA could also contribute to the lower transfection efficiencies, as the mRNA could be less readily available for translation in the cytoplasm. This statement may be valid until 4 hours of translation comparison between LAH4-L1 and LAH4-L1-R (Figure 3d). This tighter packaging might also protect the mRNA from rapid degradation by intracellular RNases, thereby contributing to the efficient translation of LAH4-L1-R compared to LAH4-L1 from a 4 h to 24 h period (10.1080/1061186X.2018.1550646 can be cited here). The authors must also note that the eGFP-MFI also depends on the half-life of eGFP in the cytoplasm, as MFI at each time is a cumulative expression of eGFP.
The efficient protection of mRNA from extracellular and intracellular RNases by LAH4-L1-R may also be one of the reasons why LAH4-L1-R outperformed LAH4-L1 in the induction of proliferation of CD8+ T cells. Efficient protein production may contribute to efficient antigen presentation.

Round 2
Reviewer 2 Report
The authors improved the discussion and contents as suggested.